# Correlation of Taste Components with Consumer Preferences and Emotions in Chinese Mitten Crabs (*Eriocheir sinensis*): The Use of Artificial Neural Network Model

**DOI:** 10.3390/foods11244106

**Published:** 2022-12-19

**Authors:** Wei Ding, Qi Lu, Licheng Fan, Mingyu Yin, Tong Xiao, Xueqian Guo, Long Zhang, Xichang Wang

**Affiliations:** 1College of Food Science and Technology, Shanghai Ocean University, Shanghai 201306, China; 2Shanghai Engineering Research Center of Aquatic-Product Processing & Preservation, Shanghai 201306, China; 3Laboratory of Quality and Safety Risk Assessment for Aquatic Products on Storage and Preservation (Shanghai), Ministry of Agriculture and Rural Affairs, Shanghai 201306, China

**Keywords:** Chinese mitten crab, taste, consumer perception, emoji, artificial neural network (ANN)

## Abstract

This study took a consumer sensory perspective to investigate the relationship between taste components and consumers’ preferences and emotions. Abdomen meat (M), hepatopancreas (H), and gonads (G) of Chinese mitten crabs, one from Chongming, the Jianghai 21 variety (C-JH), and two from Taixing, the Jianghai 21 (T-JH) and Yangtze II varieties (T-CJ), were used to evaluate flavor quality. The results indicated that in the abdomen meat, differences in taste components were mainly shown in the content of sweet amino acids, bitter amino acids, K^+^, and Ca^2+^; M-C-JH had the highest EUC value of 9.01 g/100 g. In the hepatopancreas, bitter amino acids were all significantly higher in H-C-JH (569.52 mg/100 g) than in the other groups (*p* < 0.05). In the gonads, the umami amino acid content was significantly higher in G-T-JH than in the other groups (*p* < 0.05) (EUC values: G-T-JH > G-C-JH > G-T-CJ). Consumer sensory responses showed that different edible parts of the crab evoked different emotions, with crab meat being closely associated with positive emotions and more complex emotional expressions for the hepatopancreas and gonads. In comparison, consumers were more emotionally positive when consuming Yangtze II crab. H-C-JH evoked negative emotions due to high bitter taste intensities. Multifactor analysis (MFA) showed arginine, alanine, glycine, proline, K^+^, and Ca^2+^ were found to have a positive correlation with consumer preference; an artificial neural network model with three neurons was built with good correlation (R^2^ = 0.98). This study can provide a theoretical foundation for the breeding of Chinese mitten crabs, new insights into the river crab industry, and the consumer market.

## 1. Introduction

Chinese mitten crab (*Eriocheir sinensis*) is a unique and famous aquatic product in China [1]. National freshwater aquaculture production in China in 2020 was 30.89 million tons, marking an increase of 2.49% compared with 2019, of which aquaculture production of Chinese mitten crab accounted for nearly 800,000 tons. Jiangsu and Shanghai have served as the major farming bases for Chinese mitten crab [2]. Aquatic researchers have employed different techniques for breeding new varieties to improve production and quality, due to the overexploitation of natural resources and confusion of crab seedlings [3,4]. In the Yangtze River system, Jianghai 21 (MOA registration number: GS-02-003-2015) and Yangtze II (MOA registration number: GS-01-004-2013) are currently the main varieties for breeding. Jianghai 21 has a significant advantage in terms of body weight and foot length [5]. In contrast, a greater focus was placed on growth rate and individual size in Yangtze II. Crab breeding has been developed for over 30 years, whereas there have been few studies on the differences in taste quality of different varieties.

Chinese mitten crab is beloved for its umami and sweet taste, derived from taste compounds including nitrogenous (free amino acids (FAAs) and nucleotides) and non-nitrogenous compounds (inorganic ions) [6]. Existing research on the flavor quality of Chinese mitten crab has primarily focused on meat. Arginine (Arg), glycine (Gly), alanine (Ala), glutamic acid (Glu), inosine monophosphate (IMP), and adenosine monophosphate (AMP) are of great significance due to their high taste activity value (TAV) in crab meat [7]. A study on the effect of temporary rearing in brackish water on the flavor quality of cooked crab meat indicated that crab body meat showed a significant increase in content of IMP, AMP, Ca^2+^, and Cl^−^ after 4 weeks of short-term rearing in brackish water, resulting in a better performance of umami taste [8]. In a study on hepatopancreas browning of the Chinese mitten crab, it was found that the content of total bile acids and bitter FAAs might be the contributors to the strong bitter taste of the dark hepatopancreas [9]. Fan et al. [10] found that FAAs and 5′-nucleotide content of the gonads from the Chinese mitten crab increased with freezing time, which was effective in distinguishing the overall flavor characteristics of the gonads during freezing. Due to changes in demographics, labor force, and incomes since the beginning of the 21st century, consumers’ demand for food is not limited to nutrition; it is more diverse, especially in sensory value [11]. Existing research on Chinese mitten crab has primarily focused on determining flavor compounds, whereas the correlation between physiochemical indicators and consumers’ emotions has been ignored. When consumers have similar overall preferences for products, affective correlations can effectively differentiate them and predict consumer behavior [12,13,14]. In contrast, vocabulary-based consumer questionnaires have problems in comprehension and translation, while emojis have validity and cross-cultural shared meaning [15]. The use of emojis has been currently applied to formulated foods (e.g., probiotic fermented milk [16], gluten-free bread [17], beer [18], and dark chocolate [19]), whereas it has not been applied to aquatic products.

There is a complex correlation between flavor compound content and consumer preference. Jackman et al. [20] used the classical statistical method of a partial least squares regression (PLSR) model for the sensory acceptability of steak, and the PLSR model responded well to the variation in steak eating quality (R^2^ = 0.88). A study based on a generalized linear modelling (GLM) regression model combined with analytical parameters to predict the consumer acceptability of cooked ham resulted in an evaluation model with R^2^ = 0.43–0.88, with a validation R^2^ = 0.56 for the ham flavor attribute [21]. More recently, artificial neural network (ANN) models have increasingly served as predictive tools in research, especially in aquatic products. Fan et al. [22] used an ANN model for accurate prediction of fat oxidation in the hepatopancreas of the Chinese mitten crab based on storage temperature and time. Wang et al. [23] found that an ANN model could successfully predict quality change in frozen tilapia fillets throughout a time period from −40 °C to −8 °C with relatively small errors in contrast to the Arrhenius model. An ANN model exhibits a strong self-learning capability and can continuously improve its modeling ability during the learning process. After learning the initial input data and their relationships, it can infer the hidden relationships among unknown data, thus enabling the model to generalize and predict unknown data [24].

The relationship between the chemical composition of Chinese mitten crabs and consumer experience has not been established. Therefore, this study hypothesized that a better understanding of this relationship might provide a predictive model of consumer preference. In this study, the correlation between flavor components, consumer preferences, and emotions were investigated through sensory testing combined with the examination of FAAs, 5′-nucleotides, and metal ions to compare the flavor quality of Chinese mitten crabs from different origins and varieties. Furthermore, predictive models for consumer preferences and flavor compounds were built using an ANN.

## 2. Materials and Methods

### 2.1. Chemicals and Reagents Section

Trichloroacetic acid (TCA), perchloric acid (PCA), sodium hydroxide, and nitric acid were of analytical grade purity. Methanol, potassium dihydrogen phosphate, and dipotassium hydrogen phosphate all had high-performance liquid chromatography (HPLC) purity. Standards for 17 amino acids: aspartic acid (Asp), glutamic acid (Glu), threonine (Thr), serine (Ser), glycine (Gly), alanine (Ala), arginine (Arg), proline (Pro), valine (Val), methionine (Met), leucine (Leu), tyrosine (Tyr), phenylalanine (Phe), lysine (Lys), histidine (His), isoleucine (Ile), and cystine (Cys) originated from Sigma Aldrich Chemical Co. (St. Louis, MO, USA). Nucleotide standards for 5′-guanosine monophosphate (GMP), 5′-inosine monophosphate (IMP), 5′-adenosine monophosphate (AMP), hypoxanthine (Hx), and inosine (HxR) were provided by Shanghai Yuanye Bio-Technology Co. (Shanghai, China). A multi-element mixture standard storage solution (100 μg/mL) containing sodium (Na^+^), potassium (K^+^), magnesium (Mg^2+^), and calcium (Ca^2+^) was purchased from the National Center for Analysis and Testing of Non-ferrous Metal and Electronic Materials (Beijing, China).

### 2.2. Samples and Preparations

The Jianghai 21 (JH) variety of Chinese mitten crabs were bred in Chongming (C) (Chongdong farming, Shanghai, China). The Jianghai 21 and Yangtze II (CJ) varieties of Chinese mitten crabs were bred in Taixing (T) (Jiangyuan farming, Jiangsu, Taixing, China). The Chinese mitten crabs collected in this study were all females; the number of crabs in each group was 100. Appendix A lists information about the rearing conditions and yields of the Chinese mitten crabs in this study.

All the crabs were caught in November 2021. After being caught, the crabs were immediately tied with twine to prevent excessive energy consumption. The crabs were placed in perforated foam boxes with ice packs to ensure proper conditions and then transported to Shanghai Ocean University in 3 h. Upon arrival, the crabs were kept overnight in a temporary rearing pond with an oxygenation pump to eliminate stress response. Subsequently, the crabs were washed, and the legs and carapace were removed. Afterward, the abdomen meat (M), hepatopancreas (H), and gonads (G) were collected and sealed in polythene bags and then frozen at −40 °C for use.

### 2.3. Analysis of Flavor Compounds

#### 2.3.1. Analysis of FAAs

Total FAAs were extracted using a previously described method [25]. The 0.5 g samples (accurate to 0.0001 g) were homogenized with 15 mL of 5% TCA and ultrasonicated for 10 min in a cooling bath. The solution was then held for 2 h at 4 °C, followed by centrifugation (H1850R, Xiangyi Co., Ltd., Changsha, Hunan, China). Subsequently, 5 mL of the supernatant was taken and the pH adjusted to 2.00 ± 0.02 with 6 mol/L NaOH solution. The solution was made up to 10 mL with ultrapure water and filtered through a 0.22 μm aqueous membrane for quantitative analysis using an amino acid analyzer (L-8800, Hitachi Co., Ltd., Tokyo, Japan).

#### 2.3.2. Analysis of Flavor Nucleotides

The 5′-nucleotides of the crabs were determined by a previously described method [26,27]. An approximately 3.0 g sample was mixed with 10 mL of 10% PCA and homogenized at 10,000 rpm for 1 min, followed by centrifugation at 10,614× *g* for 15 min at 4 °C. All supernatants were combined and the pH adjusted to 5.75 ± 0.02 using 1 mol/L and 6 mol/L KOH, filtered through a syringe filter (0.22 μm). The samples were analyzed using high-performance liquid chromatography (HPLC) with an Intersil ODS-3 C18 column (4.6 mm × 250 mm, 5 μm, GL Sciences Inc., Tokyo, Japan). The column temperature condition was 30 °C. Eluent A was methanol; eluent B was a mixture of 20 mmol/L dipotassium hydrogen phosphate and 20 mmol/L potassium dihydrogen phosphate (1:1, *v*/*v*). The flow rate was 1.0 mL/min, and UV detection was at 245 nm [28].

#### 2.3.3. Analysis of Metal Ions

The metal ions were analyzed using a previously published method [29]. A sample amount of 0.50 g (accurate to 0.001 g) was added to 5 mL of concentrated nitric acid to soak overnight with the lid was tightened. After soaked, it was put into the microwave apparatus (CEM-MARS 6 microwave digestion apparatus, CEM Corporation, Boston, MA, USA) for digestion. The digestion tank was kept at room temperature for cooling and acid draining. The sample solution was made up to the volume of 50 mL with ultrapure water. The blank was performed simultaneously. Finally, K^+^, Na^+^, Mg^2+^, and Ca^2+^ were determined using an inductively coupled plasma mass spectrometer (ICAP Q inductively coupled plasma mass spectrometer, Thermo Fisher Scientific Inc., Waltham, MA, USA).

#### 2.3.4. TAV and EUC

The TAV was obtained as the ratio of the concentration of the non-volatile flavor components that were examined in the sample to its threshold value, which is expressed as follows:(1)TAV=C/T
where *C* denotes the absolute content of the flavor substance (mg/100 g) and *T* represents the threshold value of the flavor substance (mg/100 g) [30].

The EUC value (g MSG/100 g) indicates the intensity of umami produced by the synergistic action of a mixture of umami amino acids and 5′-nucleotides, compared with a single concentration of MSG [31]:(2)$$\mathrm{EUC}=\sum a_{i}b_{i}+1218(\sum a_{i}b_{i})(\sum a_{j}b_{j})$$
where a_i_ denotes the content of umami amino acids (Glu, Asp) (g/100 g); b_i_ represents relative umami concentration for each umami amino acid to MSG (Glu = 1, Asp = 0.077); a_j_ expresses the content of 5′-nucleotides (IMP, GMP, AMP) (g/100 g); b_j_ is relative umami concentration of 5′-nucleotides to IMP (IMP = 1, GMP = 2.3, AMP = 0.18); and 1218 is a synergistic constant based on the concentration of g/100 mL used.

### 2.4. Consumer Sensory Evaluation

The samples were placed in an aluminum tray and then steamed at 100 °C for 20 min using an induction oven (IH13E9, Supor Group Co., Ltd., Yuhuan, China). Steamed edible parts of samples were assigned to polyethylene tasting cups, which were coded by three random numbers. To avoid the effect of temperature drops causing changes in flavor, the cups were kept at approximately 45 °C in a water bath before distribution to consumers.

Consumer sensory evaluation was performed in a sensory laboratory with temperature control (20 °C−25 °C), equipped with separate compartments and white light. A total of nine samples from different edible parts of the crab were examined by 93 untrained consumers. All participants (aged 19 to 34 years, 37 males and 56 females) were recruited from postgraduate students at Shanghai Ocean University with no allergies to Chinese mitten crab. Before testing, consumers were briefly informed about the sensory process. Appendix A lists detailed information regarding the evaluators.

Consumers were informed to randomly evaluate samples and cleaned their mouths with water to avoid interaction. Three tests were applied: (1) Consumers were asked to evaluate the hedonic attributes of the samples using a 9-point scale, in which 1 represents disliked extremely, 5 represents neither liked nor disliked, and 9 represents liked extremely [32]. (2) The intensity of sweetness, umami, saltiness, and bitterness was rated using a 9-point scale. (3) Check-all-that-apply (CATA) was conducted to describe the consumers’ emotions during the tasting. Based on previous research [16,33], 23 emojis to choose amongwere listed, including 13 positive emojis (
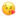
, 
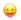
, 
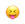
, 
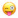
, 
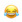
, 
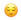
, 
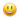
, 
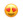
, 
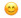
, 
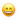
, 
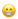
, 
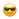
, 
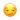
), 8 negative emojis (
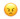
, 
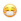
, 
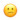
, 
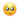
, 
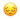
, 
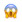
, 
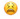
, 
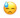
), and 2 neutral emojis (
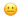
, 
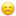
). Sensory results were collected using an electronic questionnaire (www.wenjuan.com (accessed on 16 November 2021)), and consumers completed the content via mobile phone. The study was approved by the ethical committee of Shanghai Ocean University (SHOU-DW-2021-085).

### 2.5. Data Processing

The results were shown as the mean and standard deviation. Analysis of variance (ANOVA) was performed using SPSS 26.0 (SPSS Inc., Chicago, IL, USA) at a significance level of 0.05 to evaluate whether the differences among the mean values of the different groups were statistically significant or not [34].

The choice of emoji from the consumer sensory response was transformed into the frequency of each emoji and analyzed under the guidance of the CATA approach [35]. The Cochran test was conducted using the McNemar (Bonferroni) multiple comparison test. Correspondence analysis (CA) was conducted to examine the relationship between the samples and emoji which consumers had chosen in the CATA questionnaire, Chi-square distance was chosen to measure the variability between variables and the significance level was 0.05.

Multifactor analysis (MFA) and vector regression (VR) was carried out to verify the relationship between flavor compounds (free amino acids, 5′-nucleotides and metal ions) and all sensory variables (all perceived intensity of flavor, enjoyment intensity, and CATA-based facial emoji) with samples. In the MFA, the data types were mixed.

The CATA data analysis and MFA were performed using XLSTAT 2021 software (Addinsoft, New York, NY, USA).

The ANN predictive modeling was performed using MATLAB (R 2020a) with Arg, Ala, Gly, Pro, K^+^, and Ca^2+^ as the input variables, and with consumer sensory preferences as the output variables. The proportions of the training, validation, and test sets were 70%, 15%, and 15%, respectively. The input and output variables in this model were normalized based on their possible ranges to avoid data saturation and reduce numerical difference while increasing the convergence speed of the network and the stability of the model using the following equation [36]:(3)$$A_{\mathrm{norm}}{=(a\;-\;a}_{\min})/{(a}_{\max}{\;-\;a}_{\min})$$
where *a*, *a_min_*, *a_max_*, and *a_norm_* denote the the sample data value, the minimum and maximum values of the input variable, and its normalized value, respectively.

The network based on VR correlation was visualized using Cytoscape software (version 3.9.1, Cytoscape team, Seattle, WA, USA).

## 3. Results and Discussion

### 3.1. Identification of Flavor Compounds

Flavor quality may drive consumer preference and purchase intention (including FAAs, 5’-nucleotides, and metal ions) [7,37]. The flavor evaluation of the Chinese mitten crab is shown in Table 1 and Figure 1.

#### 3.1.1. FAAs and Their TAV

FAAs were classified into umami (Glu and Asp), sweet (Thr, Ser, Gly, Ala, and Pro), and bitter (Val, Met, Leu, Tyr, Phe, Lys, His, and Ile) amino acids in accordance with their flavor attribution [38]. A total of 17 FAAs were obtained in different edible parts of the Chinese mitten crabs. The abdomen meat achieved the highest total FAA content of 2435.35~2592.70 mg/100 g, followed by the hepatopancreas with 1145.47 mg/100 g. Gonads had the lowest concentration of total FAA content, which was only 559.47~631.92 mg/100 g. A previous study was consistent with our results [10].

In the abdomen meat, M-C-JH and M-T-CJ had similar umami FAA contents (73.24 ± 4.18 mg/100 g, 75.21 ± 8.45 mg/100 g, respectively), significantly higher than that in M-T-JH (57.29 ± 5.60 mg/100 g) (*p* < 0.05). The Yangtze II crab variety contained 60.04% more sweet FAAs than the Jianghai 21 variety. There were significant differences in the crabs from different origins for bitter FAAs (*p* < 0.05). As depicted in Figure 1, there were eight types of FAAs with TAV > 1 (including Glu, Gly, Ala, Arg, Pro, Met, Lys, and His), half of which were sweet FAAs, suggesting that sweetness is the dominant flavor attribute of the abdomen meat [8,26]. Gly affects the immune system and has anti-inflammation properties, while also providing a nitrogen source for other amino acids, thus affecting the content of other FAAs [39]. Arg, with the highest TAV (12.89–13.47), has a sweet and bitter taste contribution, thus improving the umami taste and mouthfeel of Chinese mitten crab [40]. However, the presence of NaCl, Glu, and AMP reduce Arg’s contribution of bitterness to aquatic products [41]. Squid cultured at pH 7.0–9.0 facilitates Arg production while inhibiting it at pH 4.0–6.0, especially at pH 5.5 [42]. In this study, the rearing conditions of the Jianghai 21 variety river crab was at a pH of 7–8, while that of the Yangtze II variety had a pH of 6.76 (Appendix A), which might have caused the lower content of Arg in M-T-CJ than the other two groups.

In the hepatopancreas, H-C-JH contained 29.92% more umami FAAs than H-T-JH and 32.66% more umami FAAs than H-T-CJ due to a higher content of Asp and Glu. The percentage of bitter FAAs in H-C-JH, H-T-JH, and H-T-CJ was significantly higher than in the other two edible parts (*p* < 0.05), reaching 37.66%, 35.20%, and 31.33%, respectively. In addition, all types of bitter FAAs in H-C-JH were significantly higher than those in crab of Taixing origin (H-T-JH and H-T-CJ). This result is consistent with previous studies [43]. The hepatopancreas regulates metabolism in crabs and is susceptible to environmental and dietary influence [44]. Along with daily artificial feed, Chinese mitten crabs also consume aquatic plants from the farming environment. Tape grass, one of the natural fodders for river crabs, can transmit and accumulate bitter compounds through the food chain when river crabs feed on it [43]. Abundant tape grass was planted in the Chongdong area, which might have caused the accumulation of bitter FAAs in the hepatopancreas of crabs from that area.

In the gonads, G-T-JH had significantly more umami FAAs than the other two groups, while G-C-JH and G-T-CJ, with different origins and varieties, did not differ significantly in umami FAA content. No significant differences were identified between individual sweet and bitter FAAs in the gonads (*p* > 0.05).

#### 3.1.2. 5′-Nucleotides, TAV, and EUC

Nucleotides are energy-supplying components in organisms and flavor components in foods. During thermal processing, ATP-associated compounds degraded 5′-nucleotides (e.g., IMP, AMP, GMP, Hx, and HxR). Some of the 5′-nucleotides (GMP, IMP, and AMP) and FAAs had synergistic effects, thus increasing the intensity of flavor [31]. Five types of 5′-nucleotides were examined in abdomen meat, and there was no significant difference among the three groups of river crabs for GMP, IMP, AMP, and Hx, except for HxR (Table 1 and Figure 1). Hx and HxR are downstream degradation products of ATP-associated compounds that present an unacceptable bitter taste [10]. In the hepatopancreas, GMP was not detected. Only Hx and HxR were higher than the detection limit in the three groups of river crabs, without significant difference (*p* > 0.05). In the gonads, the Jianghai 21 variety crab had a significantly higher content of GMP (72.17~73.79 mg/100 g), IMP (1564.26~1633.55 mg/100 g), and AMP (361.10~377.55 mg/100 g) than the Yangtze II variety (58.56 ± 1.26 mg/100 g, 1454.08 ± 33.69 mg/100 g, and 295.09 ± 23.89 mg/100 g, respectively). As a result, the gonads of the Jianghai 21 variety of Chinese mitten crab may outperform the Yangtze II variety in umami intensity.

EUC is a criterion for sensory evaluation of foods related to umami FAAs (Glu and Asp) and 5′-nucleotides (GMP, AMP, and IMP) [28]. In the abdomen meat, the low umami FAA content of M-T-JH resulted in the lowest EUC among the three groups, at 6.18 g/100 g, whereas no significant difference was identified. For gonads, EUC values ranked as G-T-JH > G-C-JH > G-T-CJ, with significant differences (*p* < 0.05), which was consistent with the results of 5′-nucleotide content testing.

#### 3.1.3. Metal Ions and Their TAV

The content and TAV of metal ions in Chinese mitten crab are shown in Table 1 and Figure 1. No significant difference was identified in the content of all metal ions in the hepatopancreas and gonads of crabs. The probable reason for this result is that inorganic ions are vital compounds in the regulation of osmotic pressure in the organism [42,45], while the hepatopancreas and gonads are the metabolic organs and reproductive organs, which do not take on a critical significance in regulating osmotic pressure. Water-soluble metal ions have a salty or bitter taste, whereas they can act synergistically with other flavor compounds. Na^+^ exhibits flavor activity in seafood; flavor extracts lacking Na^+^ are commonly characterized by a significant reduction in sweet, salty, umami, and characteristic flavors and an increase in bitterness [6]. According to the key flavor compounds in *Takifugu obscurus*, the absence of Na^+^ and K^+^ leads to a significant decrease in sweet, umami, and salty flavor in the reconstituted liquid [38]. In abdomen meat, no significant differences were identified in Na^+^ and Mg^2+^ among the three groups of the Chinese mitten crab, and the K^+^ content of M-C-JH was significantly higher than that of M-T-CJ, whereas the Ca^2+^ content of M-T-JH was significantly lower than that of M-T-CJ. For TAV, only the TAV of K^+^ > 1 indicated its significant contribution to flavor, and existing research has also suggested that K^+^ plays an important role in crustacean flavor expression.

### 3.2. Consumer Sensory Response

#### 3.2.1. Flavor Intensity

Figure 2 shows the sensory intensities of umami, sweetness, saltiness, and bitterness of three edible parts of the Chinese mitten crab. In terms of umami, the three groups of Chinese mitten crabs had an umami intensity of 4.33–5.69. From the consumers’ perspective, the highest umami intensity was identified in the abdomen meat, followed by the gonads. Notably, the EUC values for the gonads were 11.99–26.76 times higher than the abdomen meat, not consistent with the sensory results. The probable reason for the above result is the high proportion of the bitter amino acids Hx and HxR in the gonads. The above bitter components inhibited the umami of the gonads. Fan et al. [10] performed sensory evaluation of aqueous solutions of different flavor compounds (including monosodium glutamate, sucrose, quinine, and sodium chloride) with an increase of quinine concentration, while the concentration of other flavor compounds remained unchanged. Their results indicated that the sensory score of bitterness increases gradually, while the sensory scores of umami, sweet, and salty gradually decrease. The hepatopancreas had the lowest flavor intensity, whereas it was not significantly different, and H-T-JH was relatively consistent in flavor performance among consumers.

The sweetness intensity of all abdomen meat was significantly higher than that of the hepatopancreas and gonads due to the high proportion of sweet amino acids in the abdomen meat, and the quartile range of M-C-JH and M-T-JH were the same. However, the mean sweetness intensity of M-C-JH was higher than that of M-T-JH. The above results indicated that most consumers considered the sweetness intensity of M-C-JH to be higher than that of M-T-JH.

The salty intensity of M-T-JH was significantly lower than the other two groups (*p* < 0.05) at 4.22 in the abdomen meat, consistent with the results for Na*^+^* content. The quartile range of G-T-JH was smaller, and most consumers were more consistent in their evaluation of its salty taste.

For bitterness, H-C-JH had the highest bitterness intensity and was significantly higher than H-T-JH and H-T-CJ at the same site (*p* < 0.05), consistent with its highest bitter amino acid content. However, the high quartile range of H-C-JH suggests that its bitterness expression is more complex in consumer evaluations.

#### 3.2.2. Consumer Preference

As depicted in the graph of preference intensity for edible parts of Chinese mitten crabs (Figure 2E), consumers scoring 1 (extremely disliked) or 9 (extremely liked) were in the minority; most of the consumers had a preference intensity between 4 and 7, consistent with a normal distribution. Sensory response is the human response to external stimuli, the perception and experience of the nature of the material. Sensory experience is subjective and variable and is affected by the person (physiological and psychological aspects), time, and environmental conditions. Hence, the statistics often show a normal distribution [46]. The strength of preference for all groups of samples ranged from 4.69–6.30, and for all groups was higher than 5.00 except for H-C-JH. Most consumers had a positive response to Chinese mitten crabs. These results were consistent with the higher bitter amino acid content of H-C-JH, resulting in a higher bitterness intensity. It was illustrated that excessive bitterness attributes significantly reduced consumer preference, and the tape grass presence in the farming environment might account for the higher bitterness. This finding revealed that excessive tape grass in the farming environment may significantly affect consumer preference for the hepatopancreas of Chinese mitten crabs.

#### 3.2.3. Consumer Emotions

A total of 23 face emojis in this experiment were analyzed through the Cochran test and the McNemar (Bonferroni) multiple comparison test. The emojis 
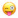
 (face with stuck-out tongue and winking eyes) and 
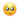
 (crying face) selected by consumers in all sample groups conveyed emotional meanings that did not correspond to the sensory response of consuming Chinese mitten crab (Table 2). Eight of the remaining 21 facial emojis significantly differentiated between the three edible parts, i.e., four positive emojis (
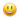
, 
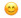
, 
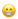
, and 
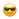
) and four negative emojis (
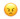
, 
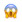
, 
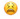
, and 
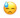
). Some researchers have suggested that specific emotional words are correlated with negative emojis, while the correlation for positive emojis is unclear [47]. The negative emojis are more likely to differ in meaning than positive emojis. The above analysis explains why four of the eight available negative emojis provided in the current experiment significantly differentiated the samples.

A correspondence analysis (CA) was then conducted, as shown in Figure 3A. The first two dimensions of the CA accounted for 74.26% of the inertia, thus illustrating the validity of the selected facial emojis. The first dimension divides the emojis in accordance with valence, with positive emojis located on the left side of the axis, neutral emojis closely distributed on the axis, and negative emojis on the right side. The second dimension accounted for 20.46% of the inertia and distinguished emojis primarily in accordance with arousal level, whose representations were not as obvious as in the first dimension [14,33,47,48]. Furthermore, the second dimension was described more by power than arousal, such that it is necessary to perform more experiments for demonstration and explanation.

Emojis with higher valence in this experiment were mainly 
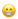
, 
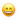
, and 
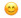
. The above three emojis were strongly correlated with happiness, with their valence ranging from 7.9–8.2, but 
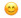
 with less activity, possibly due to the secondary correlations of embarrassment, shyness, and more complex emotions [49]. The four emojis with lower valence were 
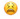
, 
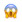
, 
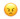
, and 
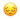
, which seems to be inconsistent with existing research, mainly in that 
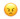
 has a higher valence than 
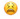
 and 
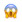
. The probable reason may be the difference in the sensory panels, as the consumers in this experiment were mainly 22- to 23-year-old Chinese students, who probably considered 
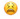
 and 
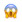
 as more negative emotions when tasting the food.

The different edible parts of Chinese mitten crab evoke various emotions, with the abdomen meat being closely related to 
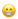
, 
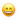
, 
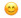
, 
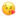
, 
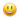
, and 
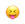
, and consumers feeling more positive when eating the Yangtze II species of crab. In contrast, the hepatopancreas and gonads are more complex in emotional expression. In the gonads, the left of the x-axis distributes Taixing G-T-CJ and G-T-JH, while the G-C-JH distribution for Chongming was to the right of the x-axis, correlated with 
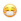
, 
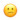
, 
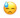
, and 
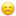
, indicating confused, unconcerned, and stressed. In the hepatopancreas, H-C-JH was more strongly correlated with 
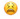
, 
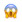
, and 
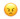
. The probable reason is that H-C-JH has a higher intensity of bitterness, which brings negative emotions to the consumer. Interestingly, the less-bitter amino acid in H-T-JH was correlated with 
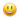
, 
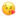
, 
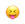
, and 
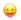
.

### 3.3. Correlation between Flavor Compounds and Consumer Sensory Preference

Vector regression (VR) of the correlations among flavor components, sensory intensity, liking, and emotion was investigated in depth through multifactorial analysis (Figure 3B). A VR coefficient between 0.6 and 0.7 inclusive indicated a moderate correlation between variables, while a VR coefficient of higher than 0.7 represented a high correlation [17]. Sensory intensity was significantly correlated with consumers’ enjoyment and emojis, moderately correlated with FAAs and metal ions, and less correlated with 5′-nucleotides. The VR coefficients for the three flavor compounds (FAAs, 5′-nucleotides, and metal ions) and consumer enjoyment were all lower than 0.6 (0.353, 0.051, and 0.186, respectively), probably due to the large composition of flavor compounds and the fact that only individual flavor compounds were correlated with preference. However, 5′-nucleotides may have slightly affected consumer preference during the comparison of the three types of flavor compounds.

Figure 3C,D visualizes the correlation among flavor components, sensory intensity, consumers’ enjoyment, and sensory emotions for the Chinese mitten crab. The first two dimensions involved 77.28% of the information. 44.35% of the data variables were explained in the first dimension, showing positive correlations with Arg, Ala, Gly, Pro, K^+^, Ca^2+^, umami intensity, sweetness intensity, enjoyment, 
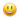
, 
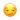
, and 
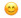
, thus distinguishing three groups of abdomen meat. In addition, AMP, IMP, GMP, Hx, HxR, Mg^2+^, 
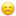
, and 
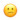
 showed negative correlations. Moreover, the above variables distinguished the three groups of gonads. The second dimension accounted for 32.94% of the data variables and showed negative correlations with Phe, Tyr, Leu, Ile, Lys, Val, bitterness intensity, 
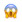
, 
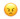
, 
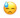
, 
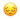
, and 
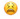
, which also significantly differentiated H-C-JH and the other two groups of hepatopancreas samples.

As a feed-forward neural network, BP neural networks are connected by neurons that allow information to flow in only one direction, from the input layer to the hidden layer and finally to the output layer [50]. From the MFA evaluation, it can be seen that preference and Arg, Ala, Gly, Pro, K^+^, and Ca^2+^ showed a high positive correlation. The number of hidden neurons was obtained using the trial-and-error method. As depicted in Table 3, the highest R^2^ (0.97072) was obtained when the number of hidden neurons was three; the mean squared error (MSE) under the same number of hidden neurons was 0.0052. Figure 4 presents the predictive performance of the neural network, thus suggesting that the preference has a good linear relationship with Arg, Ala, Gly, Pro, K^+^, and Ca^2+^. ANN models exhibit high performance in predicting consumer preferences.

## 4. Conclusions

The flavor performance of different edible parts of Chinese mitten crabs from different origins and varieties was analyzed by combining the contents of flavor components with consumer sensory evaluation. Generally, we can state conclusions as follows: (1) The flavor components of different edible parts of the Chinese mitten crabs varied with origin and variety; the Jianghai 21 variety crab bred in Chongming had more umami components in the abdomen meat and more bitter components in the hepatopancreas. (2) The emotions stimulated when consumers consumed different edible parts of the Chinese mitten crab were different. Emotion could be more positive when consumers consumed abdomen meat in contrast to the gonads and hepatopancreas of female crabs. (3) Excess bitter components could increase the intensity of bitterness perceived by consumers, leading to a reaction of negative emotions, which in turn reduces consumer preference. (4) The content of Arg, Ala, Gly, Pro, K^+^, and Ca^2+^ combined with an ANN model could be the critical flavor components for predicting consumer preference. This study can provide a theoretical foundation for the breeding of Chinese mitten crabs, and new insights into the river crab industry and the consumer market. In addition, the study has several limitations, including the diversity of the sensory panelists and sample size for modelling. In future studies, the relationship between the volatile compounds and texture of Chinese mitten crab with consumer preference will be considered.

## Figures and Tables

**Figure 1 foods-11-04106-f001:**
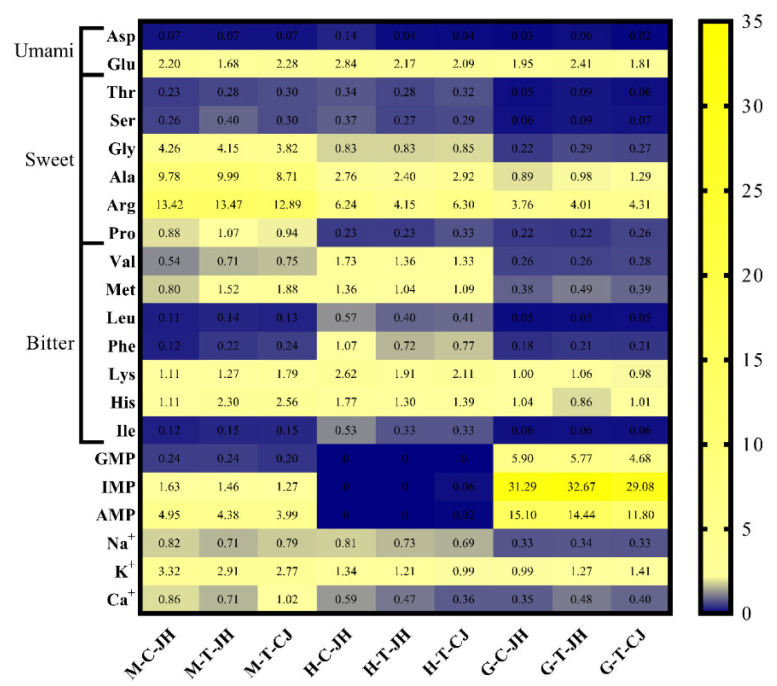
The TAV of free amino acids, 5′-nucleotides, and metal ions of Chinese mitten crab. M: abdomen meat; H: hepatopancreas; G: gonad; C: Chongming origin; T: Taixing origin; JH: Jianghai 21 variety; CJ: Yangtze II variety.

**Figure 2 foods-11-04106-f002:**
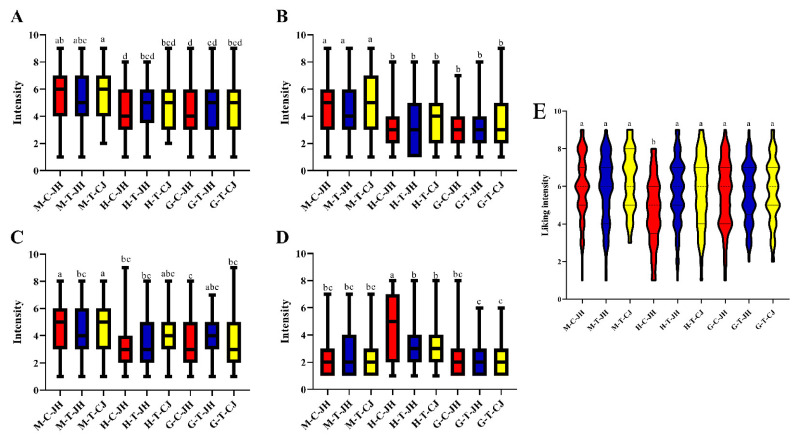
Flavor intensity and consumer preference for each edible part of the Chinese mitten crab. (**A**) umami intensity, (**B**) sweetness intensity, (**C**) saltiness intensity, (**D**) bitterness intensity, (**E**) consumer enjoyment, (*n* = 93). Different lowercase letters indicate significant differences in content (*p* < 0.05). M: abdomen meat; H: hepatopancreas; G: gonad; C: Chongming origin; T: Taixing origin; JH: Jianghai 21 variety; CJ: Yangtze II variety.

**Figure 3 foods-11-04106-f003:**
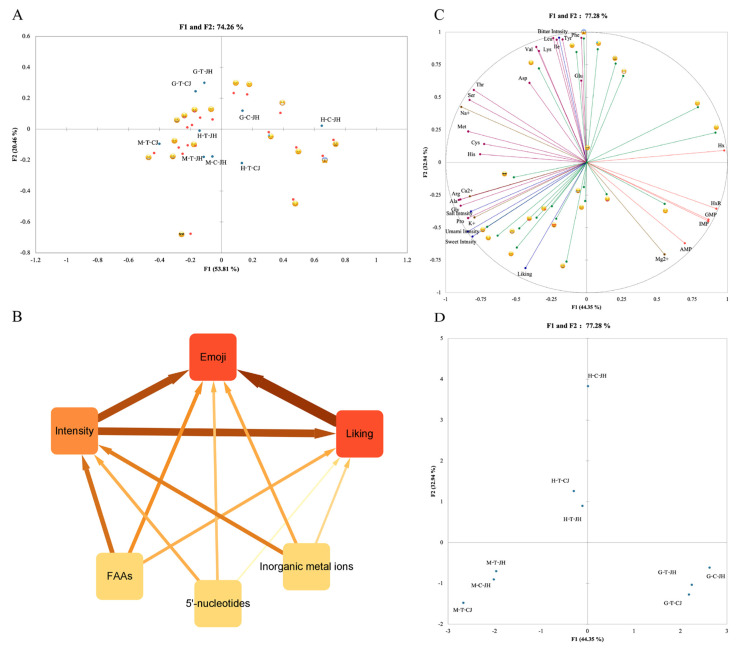
An in-depth analysis of taste components, and consumer perspectives on Chinese mitten crab. (**A**) The correspondence analysis (CA) between sample representations obtained from CATA total frequency counts (*n* = 93) with emojis. (**B**) Visual network of the flavor compound and consumers’ intensity, enjoyment, and emotion. The vector regression (VR) coefficient was visualized. The width and color shades of the line represent the size of the VR coefficient. (**C**,**D**) Relationship between flavor compounds and consumer perspectives of Chinese mitten crab based on multifactor analysis (MFA).

**Figure 4 foods-11-04106-f004:**
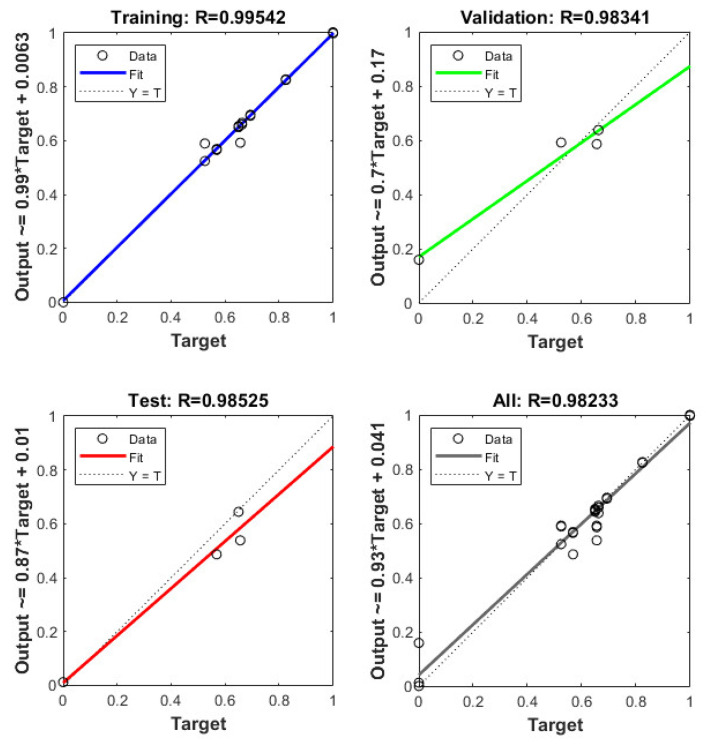
ANN model for consumer enjoyment and flavor compounds.

**Table 1 foods-11-04106-t001:** The content of free amino acids, 5′-nucleotides, metal ions, and EUC in the edible parts of Chinese mitten crabs from different origins and varieties, mg/100 g.

Name	M-C-JH	M-T-JH	M-T-CJ	H-C-JH	H-T-JH	H-T-CJ	G-C-JH	G-T-JH	G-T-CJ
Free amino acid									
Asp	7.12 ± 0.84 ^b^	6.88 ± 0.86 ^bc^	6.79 ± 0.72 ^bc^	13.55 ± 4.31 ^a^	4.10 ± 0.55 ^cde^	3.76 ± 0.13 ^de^	2.87 ± 0.71 ^e^	5.80 ± 0.14 ^bcd^	2.28 ± 0.07 ^e^
Glu	66.12 ± 3.45 ^bc^	50.41 ± 4.92 ^e^	68.42 ± 7.80 ^bc^	85.22 ± 8.31 ^a^	65.12 ± 6.94 ^bc^	62.75 ± 3.06 ^bcd^	58.39 ± 4.79 ^cde^	72.36 ± 1.67 ^b^	54.19 ± 4.60 ^de^
Thr	60.64 ± 3.44 ^c^	72.68 ± 16.6 ^abc^	79.17 ± 10.22 ^ab^	87.71 ± 11.25 ^a^	71.83 ± 7.03 ^bc^	83.14 ± 4.86 ^ab^	13.81 ± 1.16 ^d^	22.70 ± 0.40 ^d^	14.43 ± 1.10 ^d^
Ser	39.07 ± 2.45 ^b^	59.76 ± 4.84 ^a^	44.86 ± 6.57 ^b^	55.30 ± 7.02 ^a^	40.35 ± 4.29 ^b^	43.34 ± 1.31 ^b^	9.71 ± 0.83 ^c^	13.03 ± 0.17 ^c^	11.16 ± 1.13 ^c^
Gly	553.78 ± 29.75 ^a^	539.35 ± 32.65 ^ab^	496.87 ± 70.29 ^b^	108.36 ± 7.46 ^c^	108.12 ± 10.20 ^c^	110.90 ± 6.13 ^c^	28.74 ± 2.24 ^d^	37.91 ± 0.53 ^d^	35.36 ± 2.26 ^d^
Ala	586.54 ± 35.72 ^a^	599.40 ± 24.66 ^a^	522.50 ± 61.16 ^b^	165.35 ± 2.96 ^c^	143.71 ± 11.59 ^c^	174.98 ± 4.30 ^c^	53.12 ± 4.03 ^d^	58.52 ± 1.02 ^d^	77.30 ± 6.30 ^d^
Arg	671.10 ± 30.47 ^a^	673.59 ± 35.43 ^a^	644.64 ± 86.96 ^a^	311.77 ± 10.47 ^b^	207.29 ± 23.34 ^c^	314.75 ± 14.65 ^b^	188.07 ± 19.48 ^d^	200.29 ± 3.47 ^d^	215.50 ± 21.05 ^d^
Pro	264.48 ± 15.21 ^b^	321.72 ± 11.66 ^a^	282.44 ± 31.12 ^b^	68.69 ± 4.70 ^d^	69.58 ± 5.00 ^d^	97.55 ± 3.46 ^c^	66.07 ± 6.12 ^d^	65.61 ± 1.80 ^d^	79.06 ± 25.39 ^cd^
Val	21.58 ± 1.47 ^d^	28.30 ± 1.55 ^c^	29.82 ± 5.33 ^c^	69.32 ± 7.35 ^a^	54.48 ± 6.02 ^b^	53.33 ± 2.35 ^b^	10.26 ± 1.09 ^e^	10.49 ± 0.08 ^e^	11.12 ± 0.63 ^e^
Met	24.05 ± 1.95 ^d^	45.66 ± 2.52 ^b^	56.35 ± 11.02 ^a^	40.87 ± 1.00 ^b^	31.16 ± 3.21 ^c^	32.58 ± 1.64 ^c^	11.43 ± 1.50 ^e^	14.69 ± 0.10 ^e^	11.78 ± 1.85 ^e^
Leu	21.32 ± 1.12 ^c^	26.37 ± 1.06 ^c^	24.94 ± 3.49 ^c^	107.98 ± 8.91 ^a^	75.16 ± 9.69 ^b^	78.28 ± 3.79 ^b^	8.61 ± 1.26 ^d^	9.65 ± 0.48 ^d^	10.39 ± 0.61 ^d^
Tyr	15.98 ± 0.98 ^d^	22.71 ± 0.97 ^c^	26.15 ± 6.09 ^c^	81.88 ± 5.63 ^a^	58.44 ± 6.78 ^b^	58.38 ± 3.00 ^b^	14.41 ± 0.76 ^d^	14.25 ± 0.52 ^d^	14.06 ± 1.02 ^d^
Phe	11.20 ± 0.81 ^d^	19.38 ± 1.07 ^c^	21.89 ± 5.53 ^c^	96.28 ± 6.77 ^a^	64.40 ± 8.09 ^b^	69.40 ± 3.17 ^b^	16.46 ± 1.50 ^cd^	19.12 ± 0.22 ^d^	19.05 ± 1.55 ^d^
Lys	55.50 ± 3.00 ^de^	63.40 ± 0.92 ^d^	89.42 ± 14.97 ^c^	130.93 ± 7.99 ^a^	95.50 ± 12.04 ^bc^	105.29 ± 4.39 ^b^	49.89 ± 2.83 ^de^	52.76 ± 2.13 ^de^	49.01 ± 4.36 ^e^
His	22.14 ± 0.92 ^cd^	45.93 ± 1.19 ^a^	51.14 ± 10.37 ^a^	35.37 ± 4.66 ^b^	26.07 ± 2.70 ^c^	27.76 ± 2.36 ^c^	20.89 ± 2.36 ^cd^	17.20 ± 0.07 ^d^	20.10 ± 1.38 ^cd^
Ile	10.65 ± 0.67 ^c^	13.18 ± 0.65 ^c^	13.50 ± 2.86 ^c^	47.77 ± 1.83 ^a^	29.69 ± 4.11 ^b^	29.81 ± 1.49 ^b^	5.57 ± 0.58 ^d^	5.79 ± 0.52 ^d^	5.55 ± 0.50 ^d^
Cys	4.08 ± 1.01 ^a^	3.97 ± 0.46 ^a^	5.14 ± 1.23 ^a^	4.94 ± 2.41 ^a^	0.46 ± 0.29 ^b^	1.76 ± 0.14 ^b^	1.17 ± 0.11 ^b^	1.25 ± 0.11 ^b^	1.58 ± 0.24 ^b^
∑UFAA	73.24 ± 4.18 ^bc^	57.29 ± 5.60 ^de^	75.21 ± 8.45 ^b^	98.77 ± 12.62 ^a^	69.22 ± 7.48 ^bcd^	66.51 ± 3.18 ^bcde^	61.26 ± 4.75 ^cde^	78.17 ± 1.81 ^b^	56.47 ± 4.65 ^e^
∑SFAA	2175.62 ± 114.83 ^ab^	2266.51 ± 101.05 ^a^	2070.49 ± 265.02 ^b^	797.18 ± 33.58 ^c^	640.88 ± 61.28 ^c^	824.66 ± 31.55 ^c^	359.52 ± 32.15 ^e^	398.07 ± 5.66 ^e^	432.81 ± 42.93 ^e^
∑BFAA	158.36 ± 8.57 ^d^	219.27 ± 4.61 ^c^	256.85 ± 48.42 ^c^	569.52 ± 39.45 ^a^	403.75 ± 49.31 ^b^	422.25 ± 20.25 ^b^	126.08 ± 7.17 ^d^	129.27 ± 1.81 ^d^	129.28 ± 9.89 ^d^
∑FAA	2435.35 ± 129.99 ^a^	2592.70 ± 105.33 ^a^	2464.05 ± 330.77 ^a^	1511.28 ± 80.64 ^b^	1145.47 ± 121.41 ^c^	1347.76 ± 56.14 b^c^	559.47 ± 42.87 ^d^	621.44 ± 9.04 ^d^	631.92 ± 56.54 ^d^
5′-nucleotides									
GMP	2.97 ± 0.92 ^c^	3.00 ± 0.45 ^c^	2.49 ± 1.22 ^c^	N.D.	N.D.	N.D.	73.79 ± 1.72 ^a^	72.17 ± 7.18 ^a^	58.56 ± 1.26 ^b^
IMP	81.39 ± 8.02 ^c^	72.79 ± 9.96 ^c^	63.56 ± 18.65 ^c^	2.75 ± 0.02 ^d^	N.D.	N.D.	1564.26 ± 66.36 ^ab^	1633.55 ± 180.88 ^a^	1454.08 ± 33.69 ^b^
AMP	123.63 ± 16.01 ^c^	109.62 ± 22.11 ^c^	99.71 ± 33.01 ^c^	0.45 ± 0.28 ^d^	N.D.	N.D.	377.55 ± 41.03 ^a^	361.10 ± 30.00 ^a^	295.09 ± 23.89 ^b^
Hx	1.94 ± 0.83 ^c^	1.94 ± 0.12 ^c^	1.41 ± 0.75 ^c^	5.42 ± 0.08 ^b^	6.37 ± 0.45 ^b^	5.86 ± 1.68 ^b^	8.61 ± 1.61 ^a^	8.84 ± 1.27 ^a^	8.29 ± 0.84 ^a^
HxR	11.83 ± 1.50 ^cd^	12.68 ± 0.78 ^cde^	10.55 ± 3.39 ^e^	15.83 ± 1.23 ^cde^	20.01 ± 1.21 ^c^	18.47 ± 5.89 ^cd^	83.52 ± 5.90 ^a^	74.63 ± 7.77 ^b^	70.80 ± 3.19 ^b^
Inorganic metal ions									
Na^+^	146.76 ± 13.26 ^a^	127.57 ± 4.84 ^a^	142.61 ± 14.80 ^a^	146.15 ± 16.68 ^a^	131.52 ± 8.08 ^a^	124.51 ± 8.31 ^a^	59.82 ± 19.38 ^b^	61.21 ± 2.62 ^b^	59.67 ± 5.75 ^b^
K^+^	431.64 ± 68.37 ^a^	378.33 ± 14.62 ^ab^	360.51 ± 39.80 ^b^	174.83 ± 15.04 ^c^	157.65 ± 7.83 ^c^	128.33 ± 5.89 ^c^	129.03 ± 53.99 ^c^	165.08 ± 6.64 ^c^	183.69 ± 7.28 ^c^
Mg^2+^	36.45 ± 1.85 ^b^	36.12 ± 1.13 ^b^	40.46 ± 4.13 ^b^	20.84 ± 1.66 ^c^	17.17 ± 0.74 ^c^	14.52 ± 0.72 ^c^	57.24 ± 23.57 ^a^	58.66 ± 1.57 ^a^	65.97 ± 0.24 ^a^
Ca^2+^	129.41 ± 13.59 ^ab^	105.95 ± 13.38 ^bc^	152.74 ± 22.69 ^a^	89.22 ± 52.46 ^bcd^	69.97 ± 15.29 ^cd^	53.51 ± 9.84 ^d^	52.98 ± 21.51 ^d^	72.31 ± 29.63 ^cd^	59.64 ± 15.30 ^cd^
EUC	9.01 ± 1.39 ^d^	6.18 ± 1.00 ^d^	7.51 ± 3.06 ^d^	0.32 ± 0.03 ^e^	0.01 ± 0.00 ^e^	0.01 ± 0.00 ^e^	128.41 ± 6.49 ^b^	165.39 ± 18.60 ^a^	108.84 ± 11.31 ^c^

Note: EUC (g MSG/100 g). N.D.: not determined. Values of FAA content are the mean ± SD (*n* = 3). Means with different letters within a row are significantly different (*p* < 0.05). M: abdomen meat; H: hepatopancreas; G: gonad; C: Chongming origin; T: Taixing origin; JH: Jianghai 21 variety; CJ: Yangtze II variety.

**Table 2 foods-11-04106-t002:** Effect of Chinese mitten crabs of different origins and varieties on frequency (%) of use for 23 facial emojis.

Emoji	Definition	M-C-JH	M-T-JH	M-T-CJ	H-C-JH	H-T-JH	H-T-CJ	G-C-JH	G-T-JH	G-T-CJ	*p*-Value
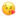	Kissing face with closed eyes	9	15	12	4	15	15	9	12	10	0.192
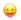	Face with stuck-out tongue	10	7	16	9	11	16	10	12	17	0.166
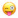	Face with stuck-out tongue and winking eyes	0	0	0	0	0	0	0	0	0	1.000
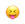	Face with stuck-out tongue and tightly closed eyes	11	8	16	9	11	8	8	11	11	0.494
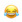	Face with tears of joy	2	8	5	3	5	7	5	7	3	0.711
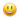	Grinning face	26 ^ab^	20 ^ab^	33 ^b^	13 ^a^	25 ^ab^	25 ^ab^	23 ^ab^	30 ^ab^	24 ^ab^	0.050
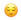	Smiling face	12	9	16	11	16	16	16	17	17	0.504
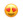	Smiling face with heart- shaped eyes	4	7	8	2	9	7	5	5	5	0.623
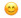	Smiling face with smiling eyes	16 ^ab^	18 ^ab^	23 ^b^	5 ^a^	17 ^ab^	18 ^ab^	13 ^ab^	16 ^ab^	13 ^ab^	0.054
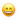	Smiling face with open mouth and tightly closed eyes	15	13	12	5	9	5	4	7	10	0.059
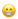	Grinning face with smiling eyes	5 ^ab^	9 ^ab^	15 ^b^	1 ^a^	11 ^ab^	7 ^ab^	7 ^ab^	7 ^ab^	4 ^ab^	0.011
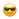	Smiling face with sunglasses	3 ^ab^	7 ^b^	1 ^ab^	0 ^b^	3 ^ab^	3 ^ab^	2 ^ab^	0 ^a^	0 ^a^	0.019
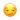	Smirking face	4	5	7	2	4	4	4	3	3	0.913
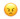	Angry face	8 ^ab^	3 ^a^	4 ^a^	15 ^b^	4 ^a^	8 ^ab^	7 ^ab^	3 ^a^	3 ^a^	0.003
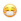	Face with medical mask	3	5	1	10	2	5	3	4	7	0.145
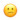	Confused face	12 ^ab^	12 ^ab^	8 ^a^	23 ^b^	14 ^ab^	12 ^ab^	19 ^b^	19 ^b^	18 ^b^	0.040
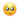	Crying face	0	0	0	0	0	0	0	0	0	1.000
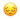	Disappointed face	5 ^ab^	3 ^a^	2 ^a^	8 ^b^	2 ^a^	8 ^b^	3 ^a^	1 ^a^	0 ^a^	0.030
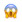	Face screaming in fear	4 a	7 ^a^	1 ^a^	17 ^b^	5 ^a^	10 ^ab^	5 ^a^	2 ^a^	3 ^a^	<0.0001
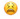	Weary face	2 ^ab^	3 ^ab^	1 ^a^	11 ^b^	1 ^a^	7 ^ab^	8 ^ab^	1 ^a^	2 ^ab^	0.001
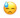	Face with cold sweat	10 ^ab^	7 ^a^	8 ^ab^	20 ^b^	14 ^ab^	10 ^ab^	13 ^ab^	7 ^a^	7 ^a^	0.017
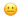	Neutral face	22	22	19	19	19	11	20	16	24	0.498
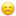	Expressionless face	16 ^a^	22 ^ab^	15 ^a^	31 ^b^	27 ^ab^	20 ^a^	33 ^b^	32 ^b^	32 ^b^	0.004

Different superscripts indicate that the frequency of the attribute for a sample differed significantly *(p* < 0.05*)*. M: abdomen meat; H: hepatopancreas; G: gonad; C: Chongming origin; T: Taixing origin; JH: Jianghai 21 variety; CJ: Yangtze II variety.

**Table 3 foods-11-04106-t003:** Changes of R^2^ and mean squared error (MSE) in the prediction of relative values under different neurons in the hidden layer.

Neurons	R^2^	MSE
1	0.37256	0.0976
2	0.62800	0.0298
3	0.97072	0.0052
4	0.59690	0.3521
5	0.42366	0.0420
6	0.42754	0.0554
7	0.85379	0.0642
8	0.53536	0.2288
9	0.65663	0.0488
10	0.42013	0.0316

## Data Availability

Data is contained within the article or Appendix A.

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
