# Peer review of "Correlation of Taste Components with Consumer Preferences and Emotions in Chinese Mitten Crabs (Eriocheir sinensis): The Use of Artificial Neural Network Model"

_foods, 2022, doi:10.3390/foods11244106_

Round 1

Reviewer 1 Report

1.Abstract should contain more result findings.

2. The Line "In this study, the correlation between taste components, consumer preferences, and emotions was be investigated through sensory test combined with the examination of  FAAs, 5'-nucleotides and metal ions to compare the taste quality of Chinese mitten crabs from different origins and varieties. Furthermore, the predictive models for consumer  preferences and taste compounds using ANN were built" Authors should elaborate why was need of such study.

3.Section 2.7 Author report Consumer Sensory analysis was done than why not trained or semi trained panelist were used?

4.Author has done extensive analysis which can be seen in the form of tables and figures.

5. As per the results and discussion presentation authors can write the conclusion in bullet point to get good understand for the study.

Author Response

Dear Editors and Reviewers:

Thank you for your letter and the comments concerning our manuscript entitled “Correlation of taste components with consumer preferences and emotions in Chinese mitten crabs (Eriocheir sinensis): the use of artificial neural network model”. (ID: foods-2408995). These comments are all valuable and very helpful for revising and improving our paper, as well as the important guiding significance to our research. We have studied comments carefully and have made correction which we hope meet with approval. Revised portions are marked in red in the paper. The main corrections in the paper and the responds to the editors and reviewers are as follow:

Responds to the reviewers’ comments:

To reviewer 1

  1. Response to comment 1: Abstract should contain more result findings.

Response: Sincerely thanks for your advice, we have added serval results into the abstract, the portion we added has been marked in red.

  1. Response to comment 2: The Line "In this study, the correlation between taste components, consumer preferences, and emotions was be investigated through sensory test combined with the examination of FAAs, 5'-nucleotides and metal ions to compare the taste quality of Chinese mitten crabs from different origins and varieties. Furthermore, the predictive models for consumer preferences and taste compounds using ANN were built" Authors should elaborate why was need of such study.

Response: Sincerely thanks for the suggestions on our articles, the comments you made were very constructive. In line 90-93 of this article, we have added the problematic points and hypothetical premises of this study.

  1. Response to comment 3: Section 2.7 Author report Consumer Sensory analysis was done than why not trained or semi trained panelist were used?

Response: Sincerely thank you for your question. First of all, our aim was to investigate the relationship between taste components with consumer preferences and emotions of Chinese mitten crab from the consumers' perspective, however, it was not practical to have all consumers undergo rigorous sensory training. After being recruited and screened consumers in the college of food science and technology at SHOU, we had a total of 100 consumers for this consumer sensory testing, all of whom had knowledge of food sensory and used to participated in food sensory testing and were given a brief step-by-step training and questionnaire prior to sensory.

  1. Response to comment 4: Author has done extensive analysis which can be seen in the form of tables and figures.

Response: Sincerely thanks for your comments.

  1. Response to comment 5: As per the results and discussion presentation authors can write the conclusion in bullet point to get good understand for the study.

Response: Sincerely thanks for your suggestions on our articles, the conclusion has been modified followed by your suggestions, which the whole conclusion was divided in four main points.

We tried our best to improve the manuscript and made some changes in the manuscript.  These changes will not influence the content and framework of the paper. And here we did not list the changes but marked in red in revised paper.

We appreciate for Editors/Reviewers’ warm work earnestly, and hope that the correction will meet with approval.

Once again, thank you very much for your comments and suggestions.

Thank you and best regards.

Yours sincerely,

Wei Ding

Xichang Wang

Ph.D. Professor

College of Food Science & Technology, Shanghai Ocean University

No.999 Hucheng Huan Road, Shanghai, P.R. China 201306

xcwang@shou.edu.cn

Reviewer 2 Report

The manuscript aims to explore the connection between taste components with consumers' preferences and emotions. The application, in three times. Analysis of variance (ANOVA), the Cochran test, the McNemar (Bonferroni) multiple comparison tests, Correspondence analysis (CA), and Multi-factor analysis (MFA) are non-well explained. Please provide, in section 2, more informations about the application of these statistical analysis.

The manuscript is not well structured. It needs more clear structure on introduction, review of literature, data and methods, empirical results and discussion, implications, and conclusion.

Abstract:

In the abstract there should more clearly stated the main aims, possible novelties and/or contributions and implications.

1. Introduction

The section on the Introduction is without clear motivation. It is suggested to specify in a better way the hypothesises, and possible novelty and/or contribution of the manuscript to the literature. Introduction should be brief, providing motivation of the research and outline main research focus. The objectives must be specified more in detail.

2. Materials and Methods

The explanations relating to the materials used for the analyses are dispersive and a little confusing. I suggest summarizing the subparagraphs from 2.1 to 2.8. Despite this, I suggest to improve the paragraph 2. For example, for the application of the ANOVA see and cite the following work:

Fanelli R.M., Di Nocera A. (2018). Customer perceptions of Japanese foods in Italy. JOURNAL OF ETHNIC FOODS, 5, 167-176. https://doi.org/10.1016/j.jef.2018.07.001

3. Results and Discussion

The results should be improved, illustrated with more emphasis and should be deepened in relation to previous studies.

How qualitative feedback can help may drive consumers’ preference and purchase intention? The discussion can improved with the following clarifications:

4. Conclusions

Conclusions should be improved as they largely repeated the results. The character of conclusion is too general. Authors should better underline conclusions, and intensions for future researches.

What are the study limitations?

What are the proposals for research in future?

Finally, Regarding Tables and Figures: the quality should be improved.

Furthermore, I suggest moving all Figures in the text for more immediate reading

Author Response

Dear Editors and Reviewers:

Thank you for your letter and the comments concerning our manuscript entitled “Correlation of taste components with consumer preferences and emotions in Chinese mitten crabs (Eriocheir sinensis): the use of artificial neural network model”. (ID: foods-2408995). These comments are all valuable and very helpful for revising and improving our paper, as well as the important guiding significance to our research. We have studied comments carefully and have made correction which we hope meet with approval. Revised portions are marked in red in the paper. The main corrections in the paper and the responds to the editors and reviewers are as follow:

Responds to the reviewers’ comments:

To reviewer 2

1.Response to comment 1: The application, in three times. Analysis of variance (ANOVA), the Cochran test, the McNemar (Bonferroni) multiple comparison tests, Correspondence analysis (CA), and Multi-factor analysis (MFA) are non-well explained. Please provide, in section 2, more information about the application of these statistical analysis.

Response: Sincerely thanks for your suggestions on our articles, data processing is a important portion for consumer sensory study. By referencing and citing relevant literature, the details of data processing have been described in section 2.5 (line 201 to 229).

2.Response to comment 2: In the abstract there should more clearly stated the main aims, possible novelties and/or contributions and implications.

Response: Sincerely thanks for your suggestions. We have modified abstract for its aim expression, novelties and potential implications to industries. The portion we modified and added has been marked in red.

3.Response to comment 3: The section on the Introduction is without clear motivation. It is suggested to specify in a better way the hypothesizes, and possible novelty and/or contribution of the manuscript to the literature. Introduction should be brief, providing motivation of the research and outline main research focus. The objectives must be specified more in detail.

Response: Sincerely thanks for your suggestions. Due to existing research on Chinese mitten crab mainly focused on taste components detection[1,2,3,4,5], whereas the relationship between taste components with consumer experience has been ignored. In addition, Chinese mitten crab is beloved for its umami and sweet taste, it means that the taste of crab would affect the consumer preference, so we set the hypothesis that a better understanding of this relation might bring up a predictive model of consumer preference. We added above in section 1(mainly in line 90 to 93) which marked in red.

[1].Zhang, L.; Yin, M.; Zheng, Y.; Xu, C.; Tao, N.; Wu, X.; Wang, X. Brackish water improves the taste quality in meat of adult male Eriocheir sinensis during the postharvest temporary rearing. Food Chem. 2020, 128409. doi:10.1016/j.foodchem.2020.128409.

[2].Zhang, L.; Tao, N.; Wu, X.; Wang, X. Metabolomics of the hepatopancreas in Chinese mitten crabs (Eriocheir sinensis). Food Res Int. 2022, 152 110914. doi:10.1016/j.foodres.2021.110914.

[3].Fan, L.; Xiao, T.; Xian, C.; Ding, W.; Wang, X. Effect of short-term frozen storage on taste of gonads of female Eriocheir sinensis and the classification of taste quality combined with sensory evaluation and fuzzy logic model. Food Chem. 2022, 378 132105-05. https://doi.org/10.1016/j.foodchem.2022.132105.

[4].Chen, D.; Zhang, M. Non-volatile taste active compounds in the meat of Chinese mitten crab (Eriocheir sinensis), Food Chem. 2007, 104 1200-05. doi:10.1016/j.foodchem.2007.01.042.

[5].Wang, S., He, Y., Wang, Y., Tao, N., Wu, X., Wang, X., Qiu, W., Ma, M. Comparison of flavour qualities of three sourced Eriocheir sinensis. Food Chem. 2016, 200 24-31. doi:10.1016/j.foodchem.2015.12.093.

4.Response to comment 4: The explanations relating to the materials used for the analyses are dispersive and a little confusing. I suggest summarizing the subparagraphs from 2.1 to 2.8. Despite this, I suggest to improve the paragraph 2.

Response: Sincerely thanks for your constructive suggestions. The section 2.2 (line 115 to 128) has been modified about stance sequence for clear reading. In addition, to make the section 2 be better structured, the analysis of FAA, flavor nucleotides and so on were summarized into a whole portion (section 2.3 Analysis of taste compounds).

5.Response to comment 5: The results should be improved, illustrated with more emphasis and should be deepened in relation to previous studies.

Response: Sincerely thanks for your advice. The section 3 (Result and Discussion) has been deepened in relation to previous studies, the portion adjusted has been marked in red.

6.Response to comment 6: How qualitative feedback can help may drive consumers’ preference and purchase intention? The discussion can improved with the following clarifications:

Response: Sincerely thanks for your questions. In this study, an artificial neural network algorithm was used to develop a predictive model of consumer preferences, using the content of 6 taste substances as inputs (arginine, alanine, glycine, proline, K+, Ca2+) and consumer preferences as outputs. This model allows practitioners or researchers to identify the key taste substances that influence consumer preferences for Chinese mitten crab; practitioners and researchers can then optimize their farming techniques by focusing on these factors during the farming process, thereby increasing consumer preferences when consuming Chinese mitten crab, which in turn increases consumer desire to buy.

7.Response to comment 7: Conclusions should be improved as they largely repeated the results. The character of conclusion is too general. Authors should better underline conclusions, and intensions for future researches.What are the study limitations?What are the proposals for research in future?

Response: Sincerely thanks for your comments and suggestions. The conclusion has been adjusted and modified for refining and summarization so that it is no longer a repetition of the results and adds to the limitations of this study and the focus of future research.

8.Response to comment 8: Finally, Regarding Tables and Figures: the quality should be improved. Furthermore, I suggest moving all Figures in the text for more immediate reading.

Response: Sincerely thanks for your suggestions. All tables and figures has been reviewed again. All figures have been set in the text for immediate reading.

We tried our best to improve the manuscript and made some changes in the manuscript.  These changes will not influence the content and framework of the paper. And here we did not list the changes but marked in red in revised paper.

We appreciate for Editors/Reviewers’ warm work earnestly, and hope that the correction will meet with approval.

Once again, thank you very much for your comments and suggestions.

Thank you and best regards.

Yours sincerely,

Wei Ding

Xichang Wang

Ph.D. Professor

College of Food Science & Technology, Shanghai Ocean University

No.999 Hucheng Huan Road, Shanghai, P.R. China 201306

xcwang@shou.edu.cn

Round 2

Reviewer 2 Report

The paper, thanks the reviewer's comments, has be improved and it is more suitable for the publication on the Journal